

# Refining the sea surface identification approach for determining freeboards in the ICESat-2 sea ice products

Ron Kwok[1], Alek A. Petty[2,3], Marco Bagnardi[2,4], Nathan T. Kurtz[2], Glenn F. Cunningham[5], Alvaro Ivanoff[2,4]

[1]Polar Science Center, Applied Physics Laboratory, University of Washington, Seattle, Washington, USA
[2]Goddard Space Flight Center, Greenbelt, Maryland, USA
[3]Earth System Science Interdisciplinary Center, University of Maryland, College Park, Maryland, USA
[4]ADNET Systems, Inc., Rockville, Maryland, USA
[5]Jet Propulsion Laboratory, California Institute of Technology, Pasadena, California, USA

*Correspondence to:* Alek Petty (alek.a.petty@nasa.gov)

**Abstract.** In Release 1 and 2 of the ICESat-2 sea ice products, candidate height segments used to estimate the reference sea surface height for freeboard calculations included two surface types: specular and smooth dark leads. We found that the uncorrected photon rates, used as proxies of surface reflectance, are attenuated due to clouds resulting in the potential misclassification of sea ice as dark leads, biasing the reference sea surface height relative to those derived from the more reliable specular returns. This results in higher reference sea surface heights and lowering estimated ice freeboards. Resolution of available cloud flags from the ICESat-2 atmosphere data product are too coarse to provide useful filtering at the lead segment scale. In Release 3, we have modified the surface reference finding algorithm so that only specular leads are used. The consequence of this change can be seen in the freeboard composites of the Arctic and Southern Ocean. Broadly, coverages have decreased by ~10-20% because there are fewer leads (by excluding the dark leads), and the composite means have increased by 0-4 cm because of the use of more consistent specular leads.



## 1   Introduction

The community distribution of higher-level data products from the ICESat-2 (IS-2) observatory (Markus et al., 2017) began with the first release in May 2019 (Release 001, R001). This was followed by a second release around October 2019 (R002), and, more recently, the third and most current release (R003). These data have all been made publicly available through the National Snow and Ice Data Center (NSIDC, https://nsidc.org/data/icesat-2). New releases are created periodically (nominally every six months), each new data product release incorporates improvements from on-going in-orbit calibration of the Advanced Topographic Laser Altimeter System (ATLAS), enhancements in the processing algorithms, and issues encountered in product generation.

One of the analyzed science products (Level 3A) from the IS-2 mission is sea ice freeboard of the polar oceans, i.e., the height of the surface above the local sea level (ATL10, Kwok et al., 2019a). The ATL10 freeboard product is generated primarily to enable calculations of sea ice thickness. To calculate sea ice freeboards, an important first step is the identification of the surface returns that could be used to estimate the height of the local sea surface. Useful freeboard estimates have been produced for the analog lidars on the ICESat mission (Kwok et al., 2007; Farrell et al., 2009) and Operation IceBridge (OIB) (Kwok et al., 2012; Kurtz et al., 2013). For the ICESat lidar (Zwally et al., 2002), investigators have used estimates of reflectance and surface relief statistics (Kwok et al., 2007), lowest level filtering (Yi et al., 2011), and waveform characteristics (Farrell et al., 2009) to separate the ice and sea surface returns. Identification of the local sea surface in the Airborne Topographic Mapper (ATM) lidar on OIB (Kurtz et al., 2013) is aided by coincident and contemporaneous digital camera images and an infra-red radiometer data. However, accurate selection of sea surface samples is very much dependent on the specific instrument (e.g. resolution, sampling, incidence angle, radiometry etc.) and whether ancillary data are available in the ice-water discrimination procedure.

The ATLAS data from IS-2 is unique in that the photon height distributions from the instrument have to be treated somewhat differently even though the physical basis for freeboard calculations remain unchanged. The classification algorithm for discriminating surface type of a height segment in the IS-2 sea ice data utilizes three attributes of the photon cloud/height distribution (photon rate, width of photon distribution, and background) to determine the surface type of a height sample. From the available IS-2 surface types, two surface types (specular and smooth dark leads) are selected as candidate height samples to estimate the sea surface reference heights and a weighted sum of the heights of these two surfaces is used for freeboard calculations. This was the approach used in R001 and R002 and is based on our pre-launch understanding of the IS-2 instrument, our experience with ICESat and an airborne implementation of a multibeam experimental lidar flown between 2012 and 2014 (Kwok et al., 2014) .

With more than a year of IS-2 data now available, together with coincident data from Operation IceBridge, we are able to better understand the capabilities of the instrument and refine the sea ice algorithm. A key outcome of or initial assessments is improved understanding of the impact of clouds on the ice-water discrimination procedure. Misidentified sea surface segments can have observable impacts on freeboard determination as errors in sea surface reference heights affect freeboard estimates over the entire 10-km freeboard determination length scale, whereas ice segment height errors affect only the individual ice surface height/freeboard estimates.

Based on the results of our analysis presented here, we find that the photon rates, used as proxies of surface reflectance are predictably attenuated due to clouds, leading to incorrect classification of ice as dark leads, and reference sea surface heights from dark leads being biased relative to the heights from the more reliable specular returns. In R003, we have modified the surface reference algorithm so that only specular leads are used. The analysis, the rationale and the impact of this revision to the sea ice algorithms are the subjects of this paper.



The paper is organized as follows. Section 2 describes the two IS-2 sea ice products (heights and freeboards) and Continuous Airborne Mapping By Optical Translator (CAMBOT) imagery obtained by Operation IceBridge used here. A brief description of the key features of the height and surface type classification algorithm is provided in Section 3. Sections 4 discusses the effect of clouds on sea surface identification, a potential approach for removing of these erroneous surface type for

consideration in reference height calculations, and the implemented change in Release 003 for addressing the impact of clouds in sea surface samples. Section 5 describes the expected differences between Releases 001/002 and 003. The last section concludes the paper.

## 2   Data description

Two data sets are used here: 1) the sea ice products from IS-2, and 2) the camera images from Operation IceBridge. They

are described below.

### 2.1  ICESat-2 (ATL07 and ATL10 products)

The ATL07 product contains profiles of surface heights and surface type of individual height segments along each of the six ground tracks (Kwok et al., 2019b). Individual ATL07 height estimates are derived from height distributions constructed using a fixed aggregate of 150 geolocated photons from the ATLAS Global Geolocated Photon Data product (ATL03)

(Neumann et al., 2019). Individual ATL10 freeboard estimates are derived from the surface heights from ATL07. A local sea surface reference ($h_{ref}$) (i.e., the estimated local sea level) is derived from the heights of available lead segments (one or more) within a 10-km along-track section (for each beam). Each lead may contain one or more consecutive sea surface height (SSH) segments. The derived sea surface references are interpolated to obtain estimates between gaps of < 50 km in length and extrapolated to adjacent 10 km sections where gaps are > 50 km. Within each 10-km section, individual freeboard heights ($h_f$)

are calculated as the difference between the surface heights ($h_s$) and the local sea surface reference (i.e., $h_f = h_s - h_{ref}$). In ATL10, freeboards are provided only where the ice concentration is higher than 50% and the height samples are at least 25-km away from the coast (to avoid uncertainties in coastal tides corrections). The ATL07 and ATL10 products (currently R002 and R003) are available the National Snow and Ice Data Center (Kwok et al., 2019b).

### 2.2  CAMBOT: Operation IceBridge

We use CAMBOT imagery obtained during the spring 2019 Operation IceBridge (OIB) Arctic campaign. CAMBOT is a nadir-looking digital camera system operated by the ATM instrument team, that provides georeferenced and orthorectified imagery with a spatial resolution of ~9 cm at the nominal flight altitude of 500 m. The CAMBOT data are available through the NSIDC (https://nsidc.org/data/iocam1b). The spring 2019 OIB Arctic campaign surveyed the thicker multi-year ice north of Ellesmere Island and was designed to optimize spatial/temporal coincidence with IS-2 (see Figure 1 in Kwok et al., 2019). Winds

(and thus sea ice drift) were reported to be low throughout these flights, increasing coincidence, however the presence of leads in this highly consolidated sea ice regime was limited. Manual inspection of the CAMBOT imagery and ATL07 data identified ~10 examples of misclassified dark leads. We include two example scenes here (Section 4) which had the best spatial/temporal coincidence with IS-2. Scene 1 was obtained by CAMBOT on April 12th at 13:23:48-13:24:45 UTC (86.6N, 127.5W) which IS-2 (RGT 218, Beam 2) passed at 13:03-13:05 UTC (time difference of ~20 minutes). Scene 2 was from April 22nd at 14:07:15-

14:08:12 UTC (81.6 N, 118.2W) which IS-2 (RGT 371, Beam 2) passed at: 13:29-13:33 UTC (time difference of ~40 minutes).



## 3    Ice-water discrimination

In this section, we first provide a brief description of procedure used to separate surface types and the use of these surface types in identifying the sea surface samples used in the calculation of freeboards. Second, we show the distribution of attributes of the sea surface height samples in three months of ATL10 products (January, June, and October 2019). These three months

were chosen to broadly represent the full seasonal cycle in ATL07/10 data across both poles.

### 3.1    Identification of sea surface samples in IS-2 (in R001/02)

Each height segment in ATL07 is assigned a surface type (specular, dark_lead (smooth), dark_lead (rough), gray ice, snow-covered ice, rough, shadow). These surface types were chosen as they are expected to broadly represent the typical surfaces encountered over the polar oceans – a detailed description of the classification approach can be found in Kwok et al. (2016). The

primary use of surface types is for determining, together with local height statistics, whether a given height segment is suitable for use as a sea surface height sample in computing freeboards in ATL10. The surface type classifier uses three attributes derived from the photon distribution of a height segment, they are: photon rate ($r_{surf}$), width of photon distribution ($w_s$) and background rate ($r_{bkg}$).

The surface photon rate (photons/shot) is the average number of detected surface photons (photoelectrons) divided by the

number of laser shots required to construct a 150-photon aggregate. In the absence of clouds, it provides a measure of the brightness or apparent reflectance of the surface. Open leads of smooth open-water/thin ice surfaces at near-nadir incidence angles can be specular/quasi-specular (i.e., high photon rates) but can also have low photon rates characteristic of surfaces with low surface reflectance/albedo. Specular returns are relatively common in IS-2 sea ice returns, and these returns are especially useful as large numbers of photons over very short length scales (i.e., small number of shots with interpulse spacing of 70 cm)

are ideal for resolving very narrow leads (10s of meters) within the ice cover. Unlike the higher signal-to-noise returns from specular surfaces, the classification of low albedo surface are more prone to errors due to cloud effects (Section 4). Clouds can attenuate the strength of the surface returns because the transmitted or reflected energy are scattered away (atmospheric scattering) from the narrow field-of-view of ATLAS instrument (more on this below). Between the two extremes, the surface types are of ice/snow surfaces but may be of geophysical interest for the general understanding of surface and cloud conditions.

The Gaussian width ($w_s$) of the photon-height distribution provides a measure of the surface roughness; the width is useful in further partitioning the height segments into different surface types (e.g. a specular surface with a relatively wide Gaussian width is classified as sea ice and not a lead).

Prior to surface finding, background photons are separated from surface photons based on their distance from the mode of the height distribution (Kwok et al., 2019a). Photon events that are not classified as surface returns are designated as background

or noise photons. Background photon events could be associated with noise in the lidar instrument (e.g. stray light, detector dark counts, etc.) or scattered sunlight at the laser wavelength. Specifically, the solar background count rate ($B_s$) is the solar zenith radiance due to solar energy scattered by the surface or atmosphere and provides a useful reflectance measure for surface identification. But, the latitudinal, seasonal, and daily variability of the solar zenith makes $B_s$ more challenging to use.. Under clear skies, the surface returns from Lambertian surfaces are approximately linearly related to the solar background rate.

Deviations from a linear relationship are indicative of shadows (cloud shadows or ridge shadows), specular returns, or atmospheric scattering. In the case of quasi-specular returns from a dark lead, for example, the behavior of background vs photon rate is not positively correlated: that is, while the surface photon rate is high for quasi-specular returns, the solar background rate is low due to a low reflectance smooth surface. When the sun is up in the polar regions, the availability of solar background





provides another proxy of surface reflectance and adds to the confidence level in our surface type classification. The reader is referred to the procedure described in (Kwok et al., 2019a) for further details.

### 3.2 Post classification height filtering

When a sea surface sample is present locally, it is typically the lowest height along a height profile. Since sea surface samples designated by the classifier (specular and smooth dark leads) are not always unambiguous (i.e., subject to classification errors) and their heights are noisy estimates, the lowest point may not be the optimal estimate. In the IS-2 sea ice algorithm, we bracket the candidate samples in the surface height distribution selected to calculate our sea level reference. From the population of smooth surfaces $- H_{smooth}$ (i.e., with $w_s < 0.13m$), we define the upper and lower limits of the height bracket ($h_{UB}$, $h_{LB}$) to select the candidate samples, as follows:

1. $h_{LB}$ is the lowest height in $H_{smooth}$.

2. $h_{UB}$ is the higher of $h_{smooth}^2$ (the 2 nd-percentile in $H_{smooth}$) and ($h_{LB} + 2\sigma_e$).

$\sigma_e$ is the expected uncertainty in the retrieved surface height (~2-3 cm for smooth surfaces in IS-2 the retrieved heights). We include only the statistics of the smooth ice because we expect this represents the height range of level ice in the profile. The variable upper bound ($h_{UB}$) allows for small tilts in the sea surface along the profile such that a reasonable number of samples are included in the population used in the calculation of the sea surface; but, the height of all selected samples have to be below $h_{smooth}^2$ to remove the outliers from the classification process. For those candidate samples within these bounds, we gather up contiguous samples and label them as individual leads ($lead$(i)) such that a sea surface height can be estimated for each lead. Thus, there may be several leads within a 10-km segment and each lead may contains a variable number of sea surface samples. The rationale is that potential biases in contiguous height samples within a lead are likely correlated and would overweight sea level estimates (especially over a large lead) for a given 10-km segment; thus, separating the leads into independent samples over the 10-km span would provide a better estimate of the sea surface. For each lead, we calculate the sea surface estimate ( $\hat{h}_{lead(i)}$ ) as the weighted sum of the selected height samples ($h_i$), viz:

$$\hat{h}_{lead} = \sum_{i=1}^{N_s} \alpha_i h_i \quad and \quad \hat{\sigma}_{lead}^2 = \sum_{i=1}^{N_s} \alpha_i^2 \sigma_i^2$$

$$where \quad \alpha_i = \frac{w_i}{\sum_{i=1}^{N_s} w_i} \quad .$$

$$and \quad w_i = \exp\left(-\frac{h_i - h_{min}}{\sigma_i}\right)^2$$

$\sigma_i^2$ is the error variance of each height estimate (provided by the surface-finding routine in ATL07), $N_s$ is the number of contiguous height segments in a given lead, and $w_i$ is a weighting factor that varies with distance from $H_{lower}$.

Estimates from individual leads are then combined to obtain a sea level reference ( $\hat{h}_{ref}$ ) for a 10-km along-track section as below (weighting is based on the error variance of each lead $\sigma_{lead(i)}^2$):



$$\hat{h}_{ref} = \sum_{i=1}^{N_i} \alpha_i \hat{h}_{lead(i)}^i \quad and \quad \hat{\sigma}_{ref}^2 = \sum_{i=1}^{N_i} \alpha_i^2 \hat{\sigma}_{lead(i)}^2$$

$$where \quad \alpha_i = \frac{\frac{1}{\sigma_{lead(i)}^2}}{\sum_{j=1}^{N_i} \frac{1}{\sigma_{lead(i)}^2}}.$$

For each valid ice segment along the given beam, the freeboard and associated error variance are then given as:

$$h_f = h_i - \hat{h}_{ref} \text{ and } \sigma_f^2 = \sigma_i^2 + \hat{\sigma}_{ref}^2.$$

### 3.3 Photon rates and length of sea surface height segments

Figure 1 shows the distribution photon rates (photon/shot) and lead lengths of the sea surface height samples (strong beams). The mean photon rates of the entire height population (between ~6 – 8, Figure 1 – left panel) are dominated by the expected returns from a mixture of snow-covered sea ice of different roughness. The distributions are remarkably consistent for the three months (Jan-19, Jun-19, and Oct-19) shown here. As expected, Beam-3 has consistently weaker surface returns (transmitted energy of Beam 3 is ~0.81 of Beam-1 and -5). This is due to the lower transmitted laser energy, and thus lower return for this beam, which is consistent with pre-launch expectations and is attributable to the custom construction of the optical component used to split the laser energy into the six IS-2 beams (Neumann et al., 2019)

Because a fixed number of photons is used in surface finding, photon rates are determined by the number of shots, or along-track distance, needed to construct these 150-photon aggregates. That is, the segment length adapts to changes in photon rates from surfaces of different reflectance: height segment lengths are longer when the returns are lower and vice versa. The distributions of lead lengths (aggregate of sea surface segments described above) – used in reference height calculations – are bi-modal (Figure 1 – right panel); the modes are determined by leads that are specular/quasi-specular and by leads with very low reflectance. The lead lengths vary between ~10 m and 150 m, with modes at ~27 m (specular leads) and ~60 m (dark leads). The upper bound in segment length (~150-200 m) is controlled by a setting in the surface finding procedure that restricts the distance over which photons are aggregated over and serves to reduce the number of noise/background photons accumulated in long distance aggregates. The consequence of a longer integrating distance for estimating surface heights of dark leads are: 1) the likelihood that there is a mixture of surface types in the height segment; and, 2) the higher number of accumulated noise photons in the larger number of shots used.

For estimating the reference surface heights, the specular and dark lead heights are mixed in the weighting process above.

### 4 Effect of clouds on leads with low surface reflectance

As mentioned above, the presence of clouds reduces the surface returns (i.e., lower the photon rates) because the transmitted or reflected energy are scattered away from the field-of-view of the lidar. In this section, we illustrate the effect of clouds on the classification of low reflectance surfaces. First, we show the phenomenology in two examples from coincident IS-2 and CAMBOT observations acquired in April 2019. Second, we examine the distributions of sea surface heights in the population of specular and dark leads used in reference surface estimation. Third, we assess the fraction of the dark lead population that is likely contaminated by clouds.



### 4.1 Phenomenology

In the presence of clouds, the photon rates are unreliable proxies of brightness or apparent surface reflectance of the surface. In the first CAMBOT/IS-2 scene (Figure 2a), the attenuation effects of atmospheric moisture are evident in the coincident coverage of a 'dark' lead detected by the surface-type classifier (Figure 2a). A clear indication of the presence of clouds is the concurrent along-track decreases in IS-2 photon rate (from ~6 photons/shot to ~2 photons/shot) and increases in background rate (from ~3 MHz to 4 MHz),  followed by a recovery of both parameters to close to their expected levels. Since a dip in the recorded levels of the CAMBOT data is not seen, the clouds are likely present in the atmospheric column above the altitude of the IceBridge platform, which was ~1000 m for this flight-line. Because of the attenuated photon rates, the IS-2 samples within the linear feature (refrozen lead) in the CAMBOT image was mislabeled as a dark lead by the surface classifier. In the absence of the attenuation effects (dip in photon rates), these samples would not have been classified as a dark lead. Even though the post-classification height filter ensured that the surface height of those samples were the lowest in the neighborhood, the sampled heights are unlikely indicative of the sea surface (i.e., they are higher than the actual sea surface).

The second example shows gaps in IS-2 surface retrievals near the center of the CAMBOT image. Gaps in IS-2 data are present when the software on board the IS-2 observatory determine, by an on-board analysis of the photon density in that atmospheric column, that surface returns are unlikely to be present, thus no data are telemetered or downlinked to the ground station. This suggests the presence of clouds in the neighborhood of the gaps. In fact, large variability in photon rates and CAMBOT data is seen away from the gaps. Since this type of surface variability is unlikely of the sea ice cover in an area north of Ellesmere Island on April 22, both the IS-2 and CAMBOT data are affected by the atmosphere. Again, there is a misclassified lead near the center of the image – with a distinct dip in the surface height – even though a correct surface classification would have removed those samples as candidate sea surface segments. These two examples highlight the potential effects of clouds in surface type classification.

Why are cloud flags not used?  The crucial element in freeboard retrieval is the accurate identification of the height samples that are suitable for estimation of the local sea surface, largely because of the low density of these samples on the ice cover; and, errors in reference heights affect freeboard estimates over 10-km length scales, unlike that of the impact of errors of individual ice surface height estimates. The cloud flags in IS-2 are sampled every 400 m and not compatible with the size of the leads used here (27 – 80 m). Also, we find that the cloud flags are quite conservative: our understanding to-date is that a large number of leads would be removed if the cloud flags were used to filter the returns. The IS-2 cloud flags, as they are currently designed, are thus currently ineffective for addressing the cloud issue at the length scale of the leads in the sea ice data.

### 4.2 Sea surface height distribution of specular/dark leads

In first and second releases of the IS-2 sea ice products (R001 and R002), both specular and dark leads were used in the determination of the local (10-km) sea surface references. Here, we examine the height distributions of the population of specular and dark leads used in reference surface estimation to assess whether the distributions of dark leads introduce biases in the freeboard calculation. The height distributions of the two surface-type categories in the Arctic and Antarctic for three months in 2019 (Jan, Jun, and Oct) are shown in Figure 3.  We summarize the results as follows:

- The height distributions overlap even though the mean of the height distribution of the dark-leads are higher by up to 10 cm: the modes of the distributions are skewed relative to each other and the differences in the negative tail of the distribution are more distinct. This provides further, albeit indirect, evidence that the height distribution of the dark-leads are contaminated by incorrect classification of the surface as discussed above.





- The population of height segments classified as specular is much higher than the population classified as dark leads, except for the January 2019 Arctic distributions, meaning the overall impact and significance of the dark leads are lower.

It should also be noted that these are distributions of the sea surface height segments prior to their aggregation into leads and the weighted averaging of these segments into 10-km reference height estimates for freeboard calculations. Thus, the impact of the dark leads are further moderated in cases where there are mixture of specular and dark lead segments in a given 10-km section. The impact on monthly composites of the Arctic and Antarctic are discussed in Section 5.

### 4.3 Towards a new contrast-ratio cloud/lead filter

We have devised a simple approach to examine the fraction of dark leads that may be affected by clouds: the photon rate of a dark lead ( $PR_{lead}$ ) is compared to the height segment with the highest photon rate ( $PR_{max}$ ) in the neighborhood of the dark lead. As a simple diagnostic, we calculate the contrast ratio:

$$R = \frac{PR_{max}}{PR_{lead}}.$$

Under cloud-free and ideal conditions, we expect the contrast to be between 8 and 9, i.e., the albedo of snow-covered sea ice is >0.8 compared to the lower albedo (reflectance) of smooth open leads of ~0.1.  In less than ideal conditions (e.g. cloudy conditions), however, we expect this contrast to be lower.

Figure 4 shows the percentage of the dark-lead population with contrasts < 2, < 3, and < 4 within a ±20 km neighborhood of the dark-lead . It is evident that 70-80% of the population (for the months shown here) have a contrast ratio <4 and are potentially misclassified if clouds were not considered  in the surface-type analysis. This preliminary analysis suggests that this local contrast ratio diagnostic could provide a useful filter to address the cloud miss-classification issue. However, the efficacy of and the potential implementation of this approach needs to be examined in more detail if this were to be incorporated into the IS-2 sea ice product generation.

### 5   An algorithm revision:  R001/R002 to R003

In this section, we describe a simple revision to our current product generation algorithm – implemented for Release 003 (R003) – to eliminate the potential effects of the misclassified dark leads. Next, we show the differences between  R001/R002 and R003 in the monthly freeboard distributions and composites of the Arctic and Antarctic sea ice covers for January, June, and October of 2019.

### 5.1  Algorithm revision

In R001/R002, candidate height segments that were selected to estimate reference heights for freeboard calculations included, as discussed above, two primary surface types: specular and smooth dark leads. Given the likelihood for the mislabeling of dark lead segments as suggested by the results presented here, a simple revision to the algorithm for production of R003 has been implemented. Instead of using two surface types for reference height calculation, only the specular surface returns are used to derive the reference sea surface. This is a simple change in the software system, chosen to enable continued sea ice product generation while a more sophisticated filtering approach (as highlighted in the previous section) is tested. The overall impact of this change in the freeboard estimates is shown in the next section.





### 5.2 Differences between R002 and R003

Here, we compare the retrievals from R002 and R003 for the months of January, June, and October of 2019. The consequence of this change can be seen in the freeboard composites and distributions of the Arctic and Antarctic sea ice covers (Figures 5 and 6) and Table 1 summarizes the freeboard statistics of the distributions. The differences are summarized below:

- In the monthly composites of the Arctic and Antarctic, area coverage has decreased by ~10-20% because, by excluding the dark leads, there are fewer estimates of the local reference sea surface for freeboard calculations.
- The composite means have increased by 0-4 cm because of the use of surface heights from only specular returns in freeboard calculations. As shown in the previous section, the use of specular returns would lower the sea surface estimates and thus increase the retrieved freeboard. We also note that some of the changes are due to changes in coverage as well. The overall impact of dark leads on freeboard statistics is also dependent on the relative population of specular and dark leads. In January 2019, the two populations are comparable (Figure 3) whereas the dark-lead populations are smaller in the other months of the Arctic and Antarctic.

## 6 Conclusions

In this paper, we examine the effect of clouds on the surface-type classifier used to identify sea surface samples for determining freeboard. Based on these results, the IS-2 sea ice classification has been revised for production of Release 003 of the IS-2 ATL07 (sea ice heights) and ATL10 (freeboard) products.

In R001/R002, candidate height segments that were selected to estimate reference heights for freeboard calculations included two surface types: specular and smooth dark leads. We found that the photon rates, used as proxies of surface reflectance, are attenuated due to clouds (leading to incorrect classification of dark leads), and surface heights from dark leads are sometimes biased relative to the heights from the more reliable specular returns. This results in reference surfaces that are higher (when weighted with heights of specular leads) thus lowering the estimated freeboards. Cloud flags from ATL09 are low resolution (~400 m) and thus do not provide an effective filter at the length-scales of leads (10s of meters) detected by ICESat-2.

In R003, we revised the surface reference calculations so that only leads with specular returns are used. The consequence of the changes can be seen in the freeboard distributions composites of the Arctic Ocean and of the Antarctic. Broadly, for the three months examined here, coverages have decreased by ~10-20% because there are fewer leads (by excluding the dark leads), and the composite freeboard means have increased by 0-4 cm because of the use of surface heights from more reliable specular surfaces (i.e., closer to the local sea surface) in freeboard calculations.

### Acknowledgments

AP, MB, NK and AI carried out this work at NASA's Goddard Space Flight Center, with funding provided by the ICESat-2 Project Science Office.



**Table 1** Comparison of summary statistics and grid coverage of freeboard retrievals for the three months over the Arctic and Antarctic sea ice cover shown in Figures 5 and 6.

| meters | R002 | | R003 | |
|---|---|---|---|---|
| Arctic | Mean(S.D) | N | Mean(S.D) | N |
| Jan-19 | 0.25 (0.12) | 9908 | 0.28 (0.12) | 9096 |
| Jun-19 | 0.30 (0.14) | 8280 | 0.31 (0.14) | 8404 |
| Oct-19 | 0.22 (0.11) | 6371 | 0.24 (0.12) | 6143 |
| Antarctic | | | | |
| Jan-19 | 0.32 (0.20) | 2485 | 0.36 (0.22) | 2657 |
| Jun-19 | 0.25 (0.22) | 10961 | 0.25 (0.19) | 9985 |
| Oct-19 | 0.26 (0.20) | 6371 | 0.29 (0.21) | 6143 |





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





**Figure Captions**

Figure 1. Distributions of photon rates of all height segments and lead lengths (strong beams) in IS-2 sea ice products of the (a) Arctic and (b) Antarctic for the months of January, June, and October of 2019. Numerical values show the mode, mean and standard deviation of the distributions.

Figure 2. Effect of clouds on IS-2 photon rates, background rates and surface-type classification in IS-2 in R002 and R003 on (left) April 12, 2019 (RGT-218), and (right) April 22 (RGT-371). (top panel) ATL10 overlaid on CAMBOT RGB imagery, magenta markers indicate sea ice segments and blue indicates sea surface (smooth dark lead) in R002 ATL10; (second panel) magenta markers indicate sea ice segments in R003 (there are no lead segments); (third panel) red band intensity in the CAMBOT RBG image at the location of the ATL10 segments; (fourth panel) ATL10 surface height; (fifth

panel) ATL10 photon rate; (sixth panel) ATL10 background rate. In panels 4-6, red = R002 and black = R003. The vertical blue shading shows the location of the ATL10 sea surface reference (smooth dark lead) segment in R002. Low contrast in the CAMBOT imagery is due to low solar elevations of 8 and 11 degrees during acquistion.

Figure 3. Distribution of surface heights classified with specular (red) and dark returns (black) in the (a) Arctic and (b) Antarctic IS-2 sea ice products for the months of January, June, and October of 2019. All height segments are subject to additional

height filtering in the determination of reference surfaces used in freeboard calculations. Numerical values show the mean and standard deviation of the distributions.

Figure 4. Contrast of lead photon rate (PR-lead) with surface segment with the highest photon rate (PR-max) within ±20 km of the 'dark' lead for the months of January, June, and October of 2019. Numerical values show the number of surface height segments classified as 'dark' lead, and the percentage of population with contrast (PR-leads/PR-max) < 2.0, <3.0 and

<4.0.

Figure 5. Differences of monthly composite freeboard statistics and coverage between Releases 002 and 003 in the Arctic IS-2 sea ice products for the months of January, June, and October of 2019. Only specular leads are used in Release 003. N is the number of grid cells (25 by 25 km) that are covered and numerical values show the mean and standard deviation of the composite field.

Figure 6. As in Fig. 5 but for the Antarctic.



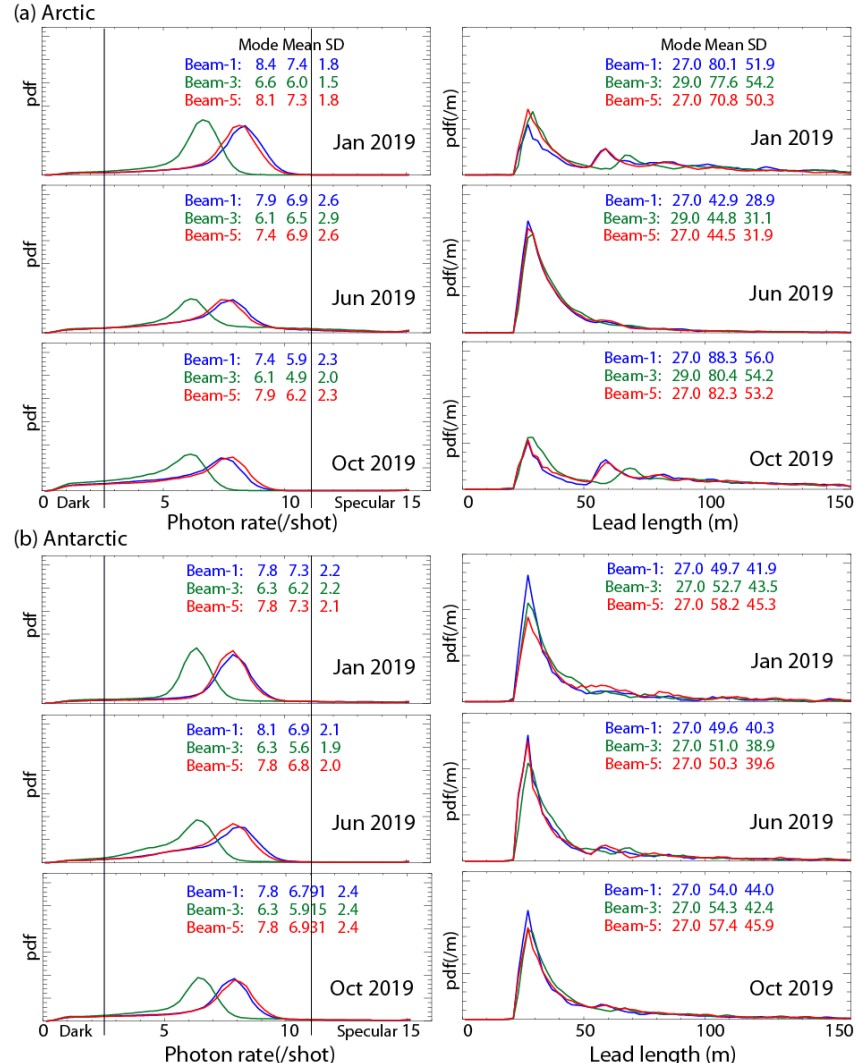

Figure 1. Distributions of photon rates of all height segments and lead lengths (strong beams) in IS-2 sea ice products of the (a) Arctic and (b) Antarctic for the months of January, June, and October of 2019. Numerical values show the mode, mean and standard deviation of the distributions.

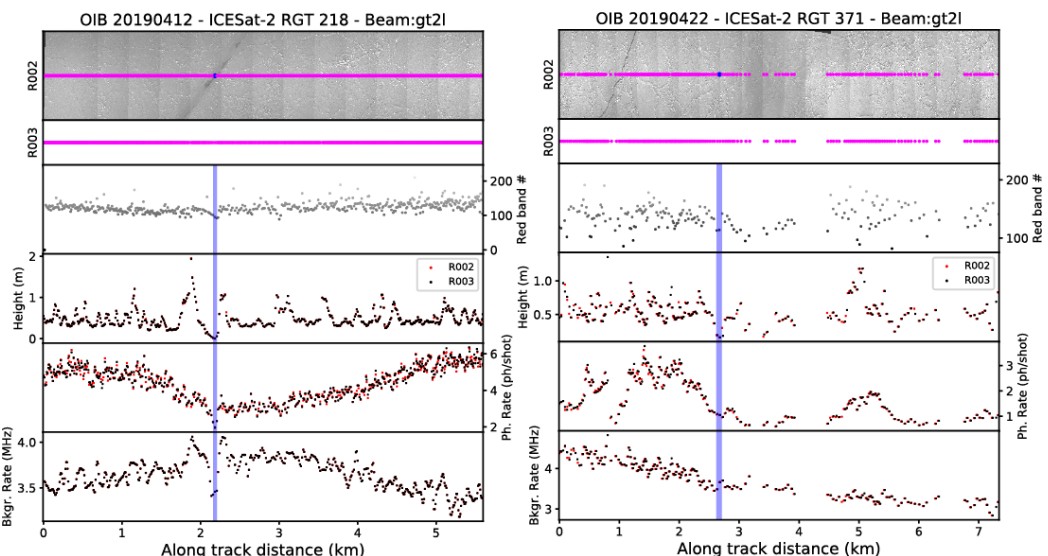

Figure 2. Effect of clouds on IS-2 photon rates, background rates and surface-type classification in IS-2 in R002 and R003 on (left) April 12, 2019 (RGT-218), and (right) April 22 (RGT-371). (top panel) ATL10 overlaid on CAMBOT RGB imagery, magenta markers indicate sea ice segments and blue indicates sea surface (smooth dark lead) in R002 ATL10; (second panel) magenta markers indicate sea ice segments in R003 (there are no lead segments); (third panel) red band intensity in the CAMBOT RBG image at the location of the ATL10 segments; (fourth panel) ATL10 surface height; (fifth panel) ATL10 photon rate; (sixth panel) ATL10 background rate. In panels 4-6, red = R002 and black = R003. The vertical blue shading shows the location of the ATL10 sea surface reference (smooth dark lead) segment in R002. Low contrast in the CAMBOT imagery is due to low solar elevations of 8 and 11 degrees during acquisition.





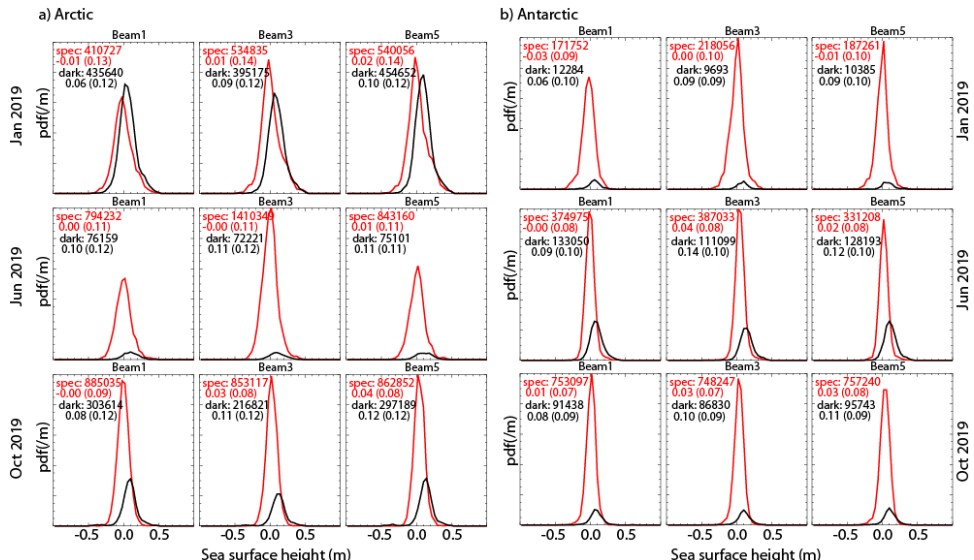

Figure 3. Distribution of surface heights classified with specular (red) and dark returns (black) in the (a) Arctic and (b) Antarctic IS-2 sea ice products for the months of January, June, and October of 2019. All height segments are subject to additional height filtering in the determination of reference surfaces used in freeboard calculations. Numerical values show the mean and standard deviation of the distributions.

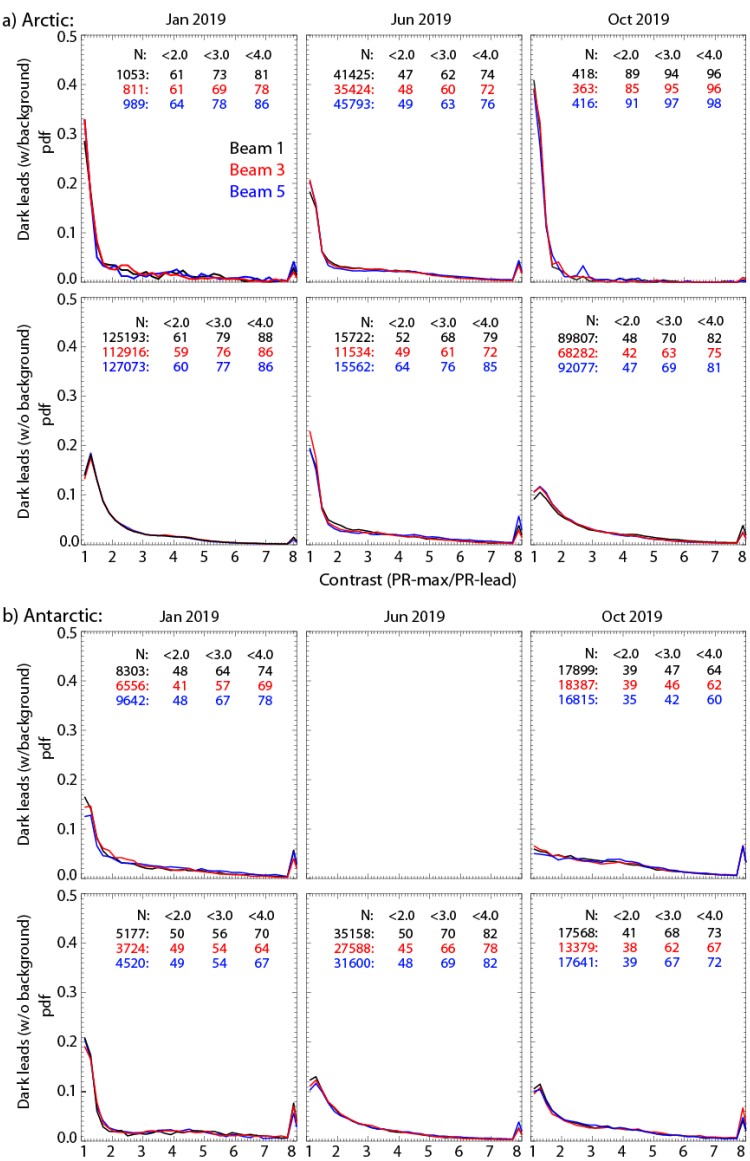

Figure 4. Contrast of lead photon rate (PR-lead) with surface segment with the highest photon rate (PR-max) within ±20 km of the 'dark' lead for the months of January, June, and October of 2019. Numerical values show the number of surface height segments classified as 'dark' lead, and the percentage of population with contrast (PR-leads/PR-max) < 2.0, <3.0 and <4.0.



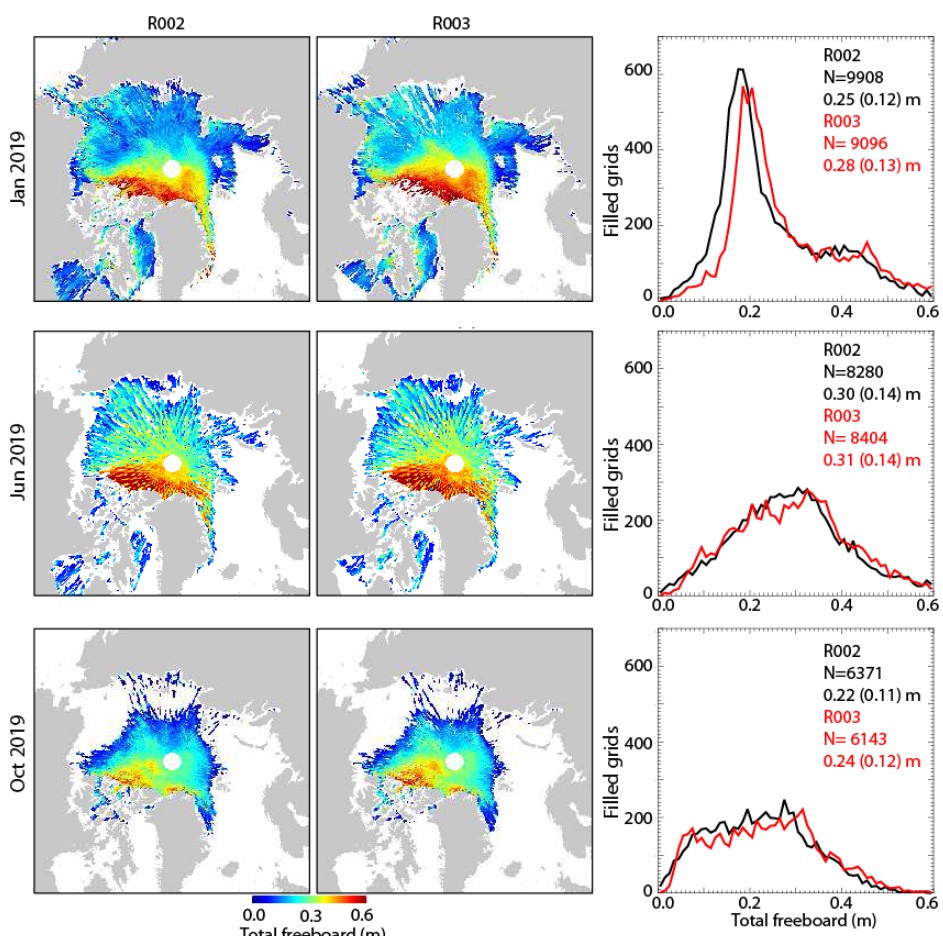

Figure 5. Differences of monthly composite freeboard statistics and coverage between Releases 002 and 003 in the Arctic IS-2 sea ice products for the months of January, June, and October of 2019. Only specular leads are used in Release 003. N is the number of grid cells (25 by 25 km) that are covered and numerical values show the mean and standard deviation of the composite field.



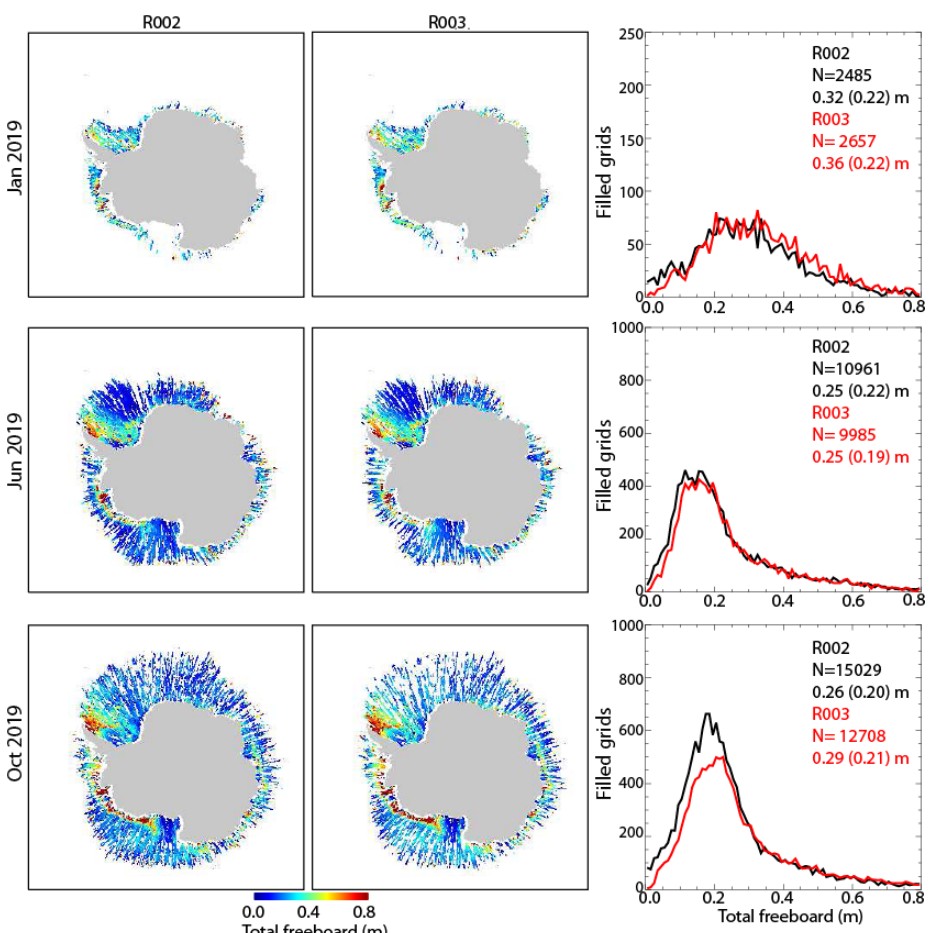

Figure 6. As in Fig. 5 but for the Antarctic