# Peer review of "Refining the sea surface identification approach for determining freeboards in the ICESat-2 sea ice products"

_The Cryosphere, 2020_

## Referee Comment (RC1) · Anonymous Referee #1 · 8 Aug 2020

This article documents the improvements of sea-surface identification with ICESat-2 altimetric measurements for sea ice, and the ensuing changes in the new release (i.e., ver. 3) of sea ice freeboard. Specifically, photon rate on dark leads can be reduced by cloud, which may cause mis-categorization as sea level tie points and result in under-estimation of sea ice freeboard. A case study with collocating airborne measurements is carried out, and a correction is introduced to utilize only specular returns for lead detection and sea-level computation. Basin-scale effects on freeboard estimation is also studied for Arctic and Antarctic retrieval. Given the importance of the ICESat-2 measurements for the ongoing and future observation of sea ice thickness, this work is an important and timely update for the polar research community. The article is also

well-written, and the presentation of the results are very good. However, I find several aspects that can be improved especially in the results and analysis part. My major comments and questions are as follows.

Major comment 1: one key analysis that I suggest to add, is the study/validation with collocating tracks of ICESat-2 and OIB. For example, the 4 OIB tracks as used in Kwok et al (2019), which showed very good consistency between OIB and IS-2 elevation, can be utilized. In fact, in Kwok et al. (2019), the effect on freeboard retrieval is discussed. But key info (including the systematic differences of freeboard, as shown Tab. 1 of the reference) may be different from the current version of the IS-2 data, including protocols, etc. Besides the basin-scale comparisons as in Sec. 5 of the manuscript, the study on the fine spatial scale may be more informative of the performance of the new treatments, as well as the consistency with OIB which feature a sea level retrieval with more comprehensive measurements. This revision is also necessary to improve the overall content of Sec. 5, which in its current form is too short and contains limited result and analysis.

Major comment 2: previous release (ver. 2) of sea ice freeboard (ATL07) contains swath-wide freeboard estimations, by using sea-level estimations across all 3 strong beams. This effectively mitigates the problem of missing parts of the beam ground segments when ssh is computed within a single beam. Besides, there also exists effect on the freeboard value for each beam segment. Since about 10% to 20% coverage is lost due to the new treatments (that excludes dark leads), this cross-beam processing could be potentially important in salvaging more data. The authors are suggested to clarify both ends of the comparison. First, whether the previous version of freeboard (R002) is based on per-beam freeboard OR swath-wide freeboard. Second, whether the new version of freeboard (R003) is based on per-beam freeboard OR swath-wide freeboard. Furthermore, whether there is a swath-wide freeboard estimation in the newly-released version of the ATL data.

Major comment 3: the discussion of the potential effect of the new scheme on sea ice

thickness is suggested to be added. During wintertime (first row in Fig. 5 and third row in Fig. 6), the average total freeboard increased by about 3cm with the new treatments. This directly translates into about 30cm systematic increase in sea ice thickness (if the snow conditions stay the same). Does this help to mitigate the observed differences between ICESat-2 thickness and CryoSat-2 estimations? [Fig. 14 of Petty et al (2020), and related text contents] The lack of detectable leads can be especially prominent within the packed ice region in the Arctic. What is the impact of losing 10% to 20% ICESat-2 measurements on the estimation of ice thickness in this region and the total volume?

Minor comments:

The analysis has been focused on strong beams of IS-2. What is the effect on the retrieval of weak beams? Does the loss of 10% to 20% of coverage include statistics from weak beams? Given the lower power of these beams (#2, #4 and #6) and the lower photon rates, is the data loss rate higher in weak beams than strong beams with the new treatments in R003?

Line 33, page 4: delete extra '.' after 'use'.

Line 11, page 5: "the second percentile in . . .".

Line 12, page 5: ". . . for smooth surfaces in IS-2 the retrieved heights" should be ". . . for smooth surfaces in the retrieved heights of IS-2".

Line 25, page 5: the definition of H_lower is missing.

Line 11, page 6: add the missing ".".

Line 17, page 8: delete the extra space.

Fig. 1 (on pg. 13): the second mode of lead width for Arctic winter by beam #3 is offset from those of beam #1 and #5 by 10 meters [top-right and bottom right sub-figure of subfigure (a)] . Why?

References:

Kwok, R., Kacimi, S., Markus, T., Kurtz, N. T., Studinger, M., Sonntag, J. G., et al. (2019). ICESat‐2 surface height and sea ice freeboard assessed with ATM lidar acquisitions from Operation IceBridge. Geophysical Research Letters, 46, 11,228–11,236. https://doi.org/10.1029/2019GL084976

Petty, A. A., Kurtz, N. T., Kwok, R., Markus, T., & Neumann, T. A. (2020). Winter Arctic sea ice thickness from ICESat‐2 freeboards. Journal of Geophysical Research: Oceans, 125, e2019JC015764. https://doi.org/10.1029/2019JC015764

---

## Referee Comment (RC2) · Anonymous Referee #2 · 22 Aug 2020

Summary

This manuscript describes a change in processing of the ICESat-2 sea ice freeboard product. It finds that returns from dark leads can be affected by clouds, giving erroneous higher surface heights, and reducing the accuracy of the freeboards. Revision 003 of the product removes dark leads as a source for the sea surface height calculation. This results in a decrease in the sea surface height and thus an increase in freeboard.

General Comment

This is nice paper that documents an important revision to the ICESat-2 sea ice freeboard product that corrects an error due to clouds in Revision 001/002 and describes the change for Revision 003 that corrects this error. The paper is well written and the analysis is complete. I have only a few minor suggestions below, the primary one being to consider moving the material in Section 4.3 to a more logical (in my view) place in the manuscript. Other comments are minor suggestions to provide more clarity in a few places.

Specific Comments (by page and line number):

P2, L33: Perhaps a bit more explanation on the 10-km freeboard determination length scale. I know this is detailed in earlier papers and the ATBD, but for a reader who is less familiar, a brief description of the method that gives rise to the 10-km length scale would be helpful. I see this noted below in Section 2.1. But still this sentence seems to presume an understanding of the products (e.g., what are "ice segment height errors"?) that hasn't yet been explicitly introduced in the manuscript. Perhaps simplify to say "Misidentified sea surface segments can have observable impacts on freeboard determine, as described below."?

P3, L2: CAMBOT is essentially a digital camera, right? I suggest adding this as a descriptor along spelling out the full name. "Optical translator" may not be clear to all readers. I see it noted in Section 2.2, but it still may be useful here. E.g., say "and digital imagery from the Continuous.... (CAMBOT) obtained by...."

P3, L12: I would provide the title of the product initially and not just the product code, e.g., "The Sea Ice Height (ATL07) product..."

P3, L25-28: Should provide recommended citation for the CAMBOT product, as noted on NSIDC's page: Studinger, M. and J. Harbeck. 2019. IceBridge CAMBOT L1B Geolocated Images, Version 2. [Indicate subset used]. Boulder, Colorado USA. NASA National Snow and Ice Data Center Distributed Active Archive Center. doi: https://doi.org/10.5067/B0HL940D452L. [Date Accessed].

P3, L34: "RGT" is introduced here and used in several other places in the manuscript, but not defined or explained. It should be spelled out the first time – Reference Ground Track – with a brief definition.

P8, L8-19: Section 4.3 seems a bit out of place here. You discuss a potential new filter and then note that you're not using that for R003, which is described in Section 5. It seems like Section 4.3 should be moved to the end as a "potential improvement for a future Revision" – i.e., as a short Section 6 (with conclusions moving to Section 7), or as a Section 5.3.

P6, L23: Last sentence of Section 3: this is/was for R001/R002, right?

Table 1: I would expect that since the dark leads are being removed in R003, the number of samples would decrease over R002. But N is larger in R003 for the Arctic Jun-19 and Antarctic Jan-19. What is the reason for this? Or am I misunderstanding what N means? (And this suggests that N should be described in the Table 1 caption.)

Figure 5: It would be useful to have a difference map for R003 minus R002. Maybe that could be the third column with the distribution plots moved to the fourth column, or if that's too crowded, maybe move the distributions to a separate figure.

Minor Comments (by page and line number):

P2, L30: typo, should be "of our" instead of "of or"

P3, L15: repetitive "from". Could say ". . .are derived from the ATL07 surface heights."

P7, L26: no hyphen in "to date"

P8, L19: spelling - "misclassification"

---

## Author Comment (AC1) · 7 Oct 2020

**Responses to Reviewer 1's comments** *(in blue):*

*This article documents the improvements of sea-surface identification with ICESat-2 altimetric measurements for sea ice, and the ensuing changes in the new release (i.e., ver. 3) of sea ice freeboard. Specifically, photon rate on dark leads can be reduced by cloud, which may cause mis-categorization as sea level tie points and result in under-estimation of sea ice freeboard. A case study with collocating airborne measurements is carried out, and a correction is introduced to utilize only specular returns for lead detection and sea-level computation. Basin-scale effects on freeboard estimation is also studied for Arctic and Antarctic retrieval. Given the importance of the ICESat-2 measurements for the ongoing and future observation of sea ice thickness, this work is an important and timely update for the polar research community. The article is also well-written, and the representation of the results are very good. However, I find several aspects that can be improved especially in the results and analysis part. My major comments and questions are as follows.*

We thank the reviewer for the valuable comments and suggested revisions.

*Major comment 1: one key analysis that I suggest to add, is the study/validation with collocating tracks of ICESat-2 and OIB. For example, the 4 OIB tracks as used in Kwok et al (2019), which showed very good consistency between OIB and IS-2 elevation, can be utilized. In fact, in Kwok et al. (2019), the effect on freeboard retrieval is discussed. But key info (including the systematic differences of freeboard, as shown Tab. 1 of the reference) may be different from the current version of the IS-2 data, including protocols, etc. Besides the basin-scale comparisons as in Sec. 5 of the manuscript, the study on the fine spatial scale may be more informative of the performance of the new treatments, as well as the consistency with OIB which feature a sea level retrieval with more comprehensive measurements. This revision is also necessary to improve the overall content of Sec. 5, which in its current form is too short and contains limited result and analysis.*

As suggested, we have repeated the analysis in Kwok et al. (2019). There were only two days where there were sea surfaces identified in the IS-2 data (April 12 and April 22). As mentioned in Kwok et al. (2019) the winds were light and the ice cover was extremely compact along these flight lines. While these conditions were ideal for acquiring coincident sea ice data sets, they were not conducive to ice deformation or open water production (i.e., leads) required for computing freeboard. Only 4 segments (10 km in length) were available for assessment of the retrieved freeboards. The results show that the IS-2 freeboards were lower than the ATM freeboards by between 3-4 cm. For the small number of samples, we did not conclude at the time that these biases were significant.

In repeating the analysis, the sea surfaces in Release 2 were not available in Release 3. This indicated that the sea surfaces used in the Kwok et al. analysis were dark leads (i.e., not classified as sea surfaces) and no longer designated as sea level references for use in freeboard calculations. This suggests that the lower IS-2 freeboards in the previous analysis may be (because of the small number of samples) due to the impact of dark leads – providing a higher (biased) sea surface and thus lower IS-2 freeboards.

*Major comment 2: previous release (ver. 2) of sea ice freeboard (ATL07) contains swath-wide freeboard estimations, by using sea-level estimations across all 3 strong beams. This effectively mitigates the problem of missing parts of the beam ground segments when ssh is computed within a single beam. Besides, there also exists effect on the freeboard value for each beam segment. Since about 10% to 20% coverage is lost due to the new treatments (that excludes dark leads), this cross-beam processing could be potentially important in salvaging more data. The authors are suggested to clarify both ends of the comparison. First, whether the previous version of freeboard (R002) is based on per-beam freeboard OR swath-wide freeboard. Second, whether the new version of freeboard (R003) is based on per-beam freeboard OR swath-wide freeboard. Furthermore, whether there is a swath-wide freeboard estimation in the newly-released version of the ATL data.*

Good point. Indeed, previous releases (vers. 1 & 2) of sea ice freeboards (in ATL10) included swath-wide freeboard estimations, by using sea-level estimations across all 3 strong beams. The swath-wide freeboard estimations should not have been and are no longer provided to users in Release 3 of ATL10 (discussed in the 'Notes to users and known issues document' – URL below). The release of the multi-beam estimates was due to an error in software implementation, which allowed the multi-beam estimates to be released. The reason for not distributing the swath-based estimates is that there are residual range biases between the heights of the three beams and until that calibration of the inter- and intra-beam range biases are available, the multi-beam biases will not be released.

https://nsidc.org/sites/nsidc.org/files/technical-references/ICESat2_ATL07_ATL10_Known_Issues_v003_Sept2020.pdf

*Major comment 3: the discussion of the potential effect of the new scheme on sea ice thickness is suggested to be added. During wintertime (first row in Fig. 5 and third row in Fig. 6), the average total freeboard increased by about 3cm with the new treatments. This directly translates into about 30cm systematic increase in sea ice thickness (if the snow conditions stay the same). Does this help to mitigate the observed differences between ICESat-2 thickness and CryoSat-2 estimations? [Fig. 14 of Petty et al (2020), and related text contents] The lack of detectable leads can be especially prominent within the packed ice region in the Arctic. What is the impact of losing 10% to 20% ICESat-2 measurements on the estimation of ice thickness in this region and the total volume?*

We are currently exploring the impact on sea ice thickness and the comparisons with CryoSat-2 thickness estimates, following on from the initial results shown in Petty et al., (2020). The reviewer is correct that the increased freeboards correspond to a significant increase in ice thickness, with the exact magnitude depending on the region and season. Our preliminary analysis across the CS-2 products suggests this pretty consistently improves the correspondence between the ICESat-2 and CryoSat-2 thickness when calculated using hydrostatic equilibrium and the same input assumptions. We choose not to show these results in this paper to keep the focus on the publicly available ICESat-2 sea ice freeboard product distributed by the ICESat-2 mission. We plan to write up the thickness comparisons in a forthcoming paper. We have yet to calculate sea ice volume, but do expect that the decreased coverage will pose a challenge, and potentially the need to introduce spatial interpolation if one was to do so.

*Minor comments:*

*The analysis has been focused on strong beams of IS-2. What is the effect on the retrieval of weak beams? Does the loss of 10% to 20% of coverage include statistics from weak beams? Given the lower power of these beams (#2, #4 and #6) and the lower photon rates, is the data loss rate higher in weak beams than strong beams with the new treatments in R003?*

The analysis is only on the strong beams. Yes, the data loss is higher. However, at this time the ICESat-2 project does not recommend the use of the weak beams because of the long integration distance needed to construct height and sea surface estimates. This is discussed in the 'known issues' document at NSIDC.

*Line 33, page 4: delete extra '.' after 'use'.*

Done.

*Line 11, page 5: "the second percentile in . . .".*
Corrected.

*Line 12, page 5: ". . . for smooth surfaces in IS-2 the retrieved heights" should be ". . . for smooth surfaces in the retrieved heights of IS-2".*
Corrected.

*Line 25, page 5: the definition of H_lower is missing.*
Corrected.

*Line 11, page 6: add the missing ".".*
Done.

*Line 17, page 8: delete the extra space.*

Done.

*Fig. 1 (on pg. 13): the second mode of lead width for Arctic winter by beam #3 is offset from those of beam #1 and #5 by 10 meters [top-right and bottom right sub-figure of subfigure (a)] . Why?*
The reason for this is that transmitted energy for Beam 3 is approximately 80% that of Beams 1 and 3. Thus, it takes a longer distance to aggregate the 150 photons used for height estimate in the surface finding procedure. This is now discussed in the text.

---

## Author Comment (AC2) · 7 Oct 2020

**Responses to Reviewer 2's comments** (*in blue):*

*This manuscript describes a change in processing of the ICESat-2 sea ice freeboard product. It finds that returns from dark leads can be affected by clouds, giving erroneous higher surface heights, and reducing the accuracy of the freeboards. Revision 003 of the product removes dark leads as a source for the sea surface height calculation. This results in a decrease in the sea surface height and thus an increase in freeboard.*

We thank the reviewer for the valuable comments and suggested revisions.

*General Comment*
*This is nice paper that documents an important revision to the ICESat-2 sea ice freeboard product that corrects an error due to clouds in Revision 001/002 and describes the change for Revision 003 that corrects this error. The paper is well written and the analysis is complete. I have only a few minor suggestions below, the primary one being to consider moving the material in Section 4.3 to a more logical (in my view) place in the manuscript. Other comments are minor suggestions to provide more clarity in a few places.*

Section 4.3 introduces a potential diagnostic for identifying dark leads that are contaminated by clouds. While moving Section 4.3 to Section 5 (or even to a new Section 6) seems more logical, we prefer to keep this subsection in Section 4 because this diagnostic has yet to be implemented and tested. In addition, the introduction of the diagnostic is closer to the discussion of the specular and dark leads. And, since Section 5 is focused on the actual algorithm revisions in Release 3, it seems more appropriate that the discussion remains in Section 4. We have added words to clarify our intent in Section 4.3 so that it does not seem out of place.

*Specific Comments (by page and line number):*

*P2, L33: Perhaps a bit more explanation on the 10-km freeboard determination length scale. I know this is detailed in earlier papers and the ATBD, but for a reader who is less familiar, a brief description of the method that gives rise to the 10-km length scale would be helpful. I see this noted below in Section 2.1. But still this sentence seems to presume an understanding of the products (e.g., what are "ice segment height errors"?) that hasn't yet been explicitly introduced in the manuscript. Perhaps simplify to say "Misidentified sea surface segments can have observable impacts on freeboard determine, as described below."?*

In the revision, we have added brief description of the 10-km length scale. And, we have used the sentence as suggested by the reviewer: *Misidentified sea surface segments can have observable impacts on freeboard determine, as described below.*

*P3, L2: CAMBOT is essentially a digital camera, right? I suggest adding this as a descriptor along spelling out the full name. "Optical translator" may not be clear to all readers. I see it noted in Section 2.2, but it still may be useful here. E.g., say "and digital imagery from the Continuous. . .. (CAMBOT) obtained by. . .."*

Correct. CAMBOT is a digital camera. The text has been revised as suggested to clarify the fact that this is a digital camera in addition to the words used in the text.

*P3, L12: I would provide the title of the product initially and not just the product code, e.g., "The Sea Ice Height (ATL07) product. . ."*

Done.

*P3, L25-28: Should provide recommended citation for the CAMBOT product, as noted on NSIDC's page: Studinger, M. and J. Harbeck. 2019. IceBridge CAMBOT L1B Geolocated Images, Version 2. [Indicate subset used]. Boulder, Colorado USA. NASA National Snow and Ice Data Center Distributed Active Archive Center. doi: https://doi.org/10.5067/B0HL940D452L. [Date Accessed]*

Done.

*P3, L34: "RGT" is introduced here and used in several other places in the manuscript, but not defined or explained. It should be spelled out the first time – Reference Ground Track – with a brief definition.*

RGT is now defined in the text.

*P8, L8-19: Section 4.3 seems a bit out of place here. You discuss a potential new filter and then note that you're not using that for R003, which is described in Section 5. It seems like Section 4.3 should be moved to the end as a "potential improvement for a future Revision" – i.e., as a short Section 6 (with conclusions moving to Section 7), or as a Section 5.3.*

See above.

*P6, L23: Last sentence of Section 3: this is/was for R001/R002, right?*

Yes, we have added this to clarify this.

*Table 1: I would expect that since the dark leads are being removed in R003, the number of samples would decrease over R002. But N is larger in R003 for the Arctic Jun-19 and Antarctic Jan-19. What is the reason for this? Or am I misunderstanding what N means? (And this suggests that N should be described in the Table 1 caption.)*

N is now defined in the text. There typos for Arctic Jun-19 and Jan-19 have been fixed.

*Figure 5: It would be useful to have a difference map for R003 minus R002. Maybe that could be the third column with the distribution plots moved to the fourth column, or if that's too crowded, maybe move the distributions to a separate figure.*

We have added difference distributions to show the impact of the revision on different intervals of freeboards.

*Minor Comments (by page and line number):*

*P2, L30: typo, should be "of our" instead of "of or"*
Corrected.
*P3, L15: repetitive "from". Could say ". . .are derived from the ATL07 surface heights."*
Done.
*P7, L26: no hyphen in "to date"*
Corrected.
*P8, L19: spelling - "misclassification"*
Corrected.

---

## Author Response (AR2)

**Responses to Editor's comments** (*in blue*):**

Both reviewers expressed their positive feedback and I am now satisfied that this paper can proceed with publication provided you incorporate the minor adjustments proposed below. I would also encourage you to give the manuscript another read to spot any remaining typos. (Lines and comments refer to the tc-2020-174-author\_response-version1.pdf document)

Thank you for serving as editor on our manuscript. Below, we document our responses to Reviewer #1 and your comments.

Major comment 1 -> please include at least some of the elements of your reply to the reviewer into the manuscript

"In repeating the analysis, the sea surfaces in Release 2 were not available in Release 3. This indicated that the sea surfaces used in the Kwok et al. analysis were dark leads (i.e., not classified as sea surfaces) and no longer designated as sea level references for use in freeboard calculations. This suggests that the lower IS-2 freeboards in the previous analysis may be (because of the small number of samples) due to the impact of dark leads – providing a higher (biased) sea surface and thus lower IS-2 freeboards consistent with our expectation of results from the algorithm revision."

The following description has been added to the end of Section 5.2:

In repeating a comparison of near-coincident freeboards from IS-2 and IceBridge in Kwok et al. (2019c), the four available sea surface references in the earlier release were not retrieved by the revised algorithm (R003). This gives an indication the sea surfaces used in the that analysis were dark leads (i.e., not classified as sea surfaces in R003) and no longer designated as sea level references (in R003) for use in ATL10 freeboard calculations. This also suggests that the lower IS-2 freeboards (compared to those from IceBridge) in that analysis may be due to the impact of dark leads, i.e., higher (biased) sea surfaces and thus lower IS-2 freeboard estimates consistent with our expectation.

Major comment 2 -> please include a reference to this into the revised manuscript. A version of your answer would do.

Minor comment -> please refer to this also (together with major comment 2)

The following description has been added to Section 2.1:

Of special note here is that, in this paper, we address only the sea surface references from individual strong beams. Previous releases (Release 1 and 2) of sea ice freeboards (in ATL10) included swath-wide (multibeam) freeboard estimates by combining sea surface references across all strong beams. Due to residual range biases (centimeter level) between the three IS-2 strong beams, these swath-wide freeboard estimations should not have been provided in ATL10. Until successful inter-beam range calibrations are satisfactorily achieved, these multibeam estimates will no longer be provided to users in upcoming releases. The release of the multi-beam estimates was due to an error in software implementation. (https://nsidc.org/sites/nsidc.org/files/technicalreferences/ICESat2\_ATL07\_ATL10\_Known\_Issues\_v003\_Nov2020.pdf)

Other edits

P1L22: clarify -> freeboard composite means Instead of '...freeboard composites...' we have re-written the sentence as follows:"... in the composites of mean freeboard of the Arctic and Southern oceans."

P4L6: description of the ...

Now reads: Identification of sea surface samples in IS-2 data (in R001/02): A brief summary

P5L18: please change naming convention of  $h^2$ \_smooth as it looks like the height is squared (maybe  $h^2$ )

Rewritten as:  $h_{smooth(2)}$ . P6L19: it takes more (remove a) Corrected.

P7L13: not sure phenomenology is the best word here

We use phenomenology to refer to the empirical observation and study of the effect of clouds in the retrieval of open leads in the data. We prefer to retain the use of the word.

P9L5: rephrase ...efficacy... Replaced with 'effectiveness'.

**Refining the sea surface identification approach for determining freeboards in the ICESat-2 sea ice products**

Ron Kwok1, Alek A. Petty2,3, Marco Bagnardi2,4, Nathan T. Kurtz2, Glenn F. Cunningham5, Alvaro Ivanoff2,4, Sahra
 Kacimi54

[revised manuscript text omitted]